# Determinants of COVID-19 Vaccination Intention among Health Care Workers in France: A Qualitative Study

**DOI:** 10.3390/vaccines10101661

**Published:** 2022-10-04

**Authors:** Coline Bourreau, Angela Baron, Michaël Schwarzinger, François Alla, Linda Cambon, Léo Donzel Godinot

**Affiliations:** 1Department of Methodology and Innovation in Prevention, Bordeaux University Hospital, 33000 Bordeaux, France; 2Inserm UMR 1219 Bordeaux Population Health, University of Bordeaux, 33000 Bordeaux, France

**Keywords:** COVID-19, healthcare workers, vaccine hesitancy, vaccination intention, SARS-CoV-2, qualitative study

## Abstract

COVID-19 vaccines are one of the best tools to limit the spread of the virus. However, vaccine hesitancy is increasing worldwide, and France is one of the most hesitant countries. From the beginning of the COVID-19 vaccination campaign, health care workers (HCW) have been prioritized in the vaccination strategy but were also hesitant. This study was conducted to identify and understand the determinants of COVID-19 vaccination intention in the French context, with a view to promoting HCW vaccination. A qualitative study using individual semi-structured interviews of HCWs was carried out at the beginning of the vaccination campaign (January to April 2021) in a French university hospital. Interviews indicated that the vaccination intention of HCWs was influenced by confidence in the proposed vaccines, past experience with vaccines and disease, the opinions and vaccination status of others, and media handling of information related to COVID-19 vaccination. Improving HCW vaccination intention regarding COVID-19 vaccines could be achieved through the dissemination of clear, reassuring, scientific information. Information should be disseminated by HCWs and vaccination experts and adapted to local contexts. To improve the level of confidence and vaccination uptake through a compliance effect, it would be useful to promote positive COVID-19 vaccination experiences and increased rates of immunization.

## 1. Introduction

Since the beginning of the SARS-CoV-2 health crisis, COVID-19 vaccines have been the most effective preventive measure in limiting the spread of the virus, relieving health care facilities and minimizing social and economic impacts [1,2,3].

From the beginning of the French vaccination campaign, health care workers (HCWs) were designated as priorities in the government’s vaccine strategy [4] since they are more likely to become infected and are potential vectors of the virus due to their professional activity. In February 2021, in French health facilities, HCWs were responsible for 34% of nosocomial SARS-CoV-2 infections [5]. COVID-19 vaccination coverage of health care workers is therefore critical in order to protect them, their patients and their families and relatives. Moreover, a population’s adherence to vaccination is mainly influenced by their opinions, immunization status and available advice [6]. HCWs have an important role to play in population immunization [7].

One of the main obstacles to COVID-19 vaccination is vaccine hesitancy [8], which is increasing worldwide, especially in France [9,10]. HCWs may also express doubts and reluctance [11]. COVID-19 vaccines have been developed rapidly, receiving emergency use authorization [12], and several COVID-19 vaccines were developed using very innovative approaches; these factors may have exacerbated vaccine hesitancy [13]. Predicted rates of COVID-19 vaccine hesitancy among caregivers are similar to those of the general population [14]. 

To promote vaccination among HCWs, exploring and understanding the determinants of their COVID-19 vaccination intention is critical for the implementation of targeted actions. In order to understand these determinants in the early phases of the vaccination campaign, we conducted semi-structured qualitative interviews among HCWs. 

## 2. Materials and Methods

### 2.1. Study Design

Our qualitative study was conducted between January and April 2021, at the start of the national COVID-19 vaccination program. Individual semi-structured interviews were used to explore and understand the perceptions and representation of HCWs about vaccination against COVID-19. The theoretical framework of the study was based on a comprehensive behavioral model of pandemic vaccination.

### 2.2. Comprehensive Behavioral Model of Pandemic Vaccination

The vaccine hesitancy model helps to explain vaccination behavior [8] and suggests various strategies to address vaccine hesitancy [15]. However, this model and related strategies do not take into account the specific context of a pandemic, such as the COVID-19 pandemic.

A comprehensive behavioral model of vaccination was developed [16]. This model aims to investigate the behavioral determinants that may be involved in pandemic vaccination (mechanism boxes in the figure). It is based on the cross-analysis of the CoVaPred study results [17], a number of meta-analyses and systematic reviews of factors involved in non-mandatory vaccination decisions (seasonal and pandemic influenza) [7,11,18,19,20,21,22] and behavior theories used for preventive measures [23,24,25,26]. 

This theory-based model identifies the key pillars (strategies boxes in the figure) most likely to promote vaccination intention in a pandemic context (Figure 1). 

### 2.3. Corpus Recruitment

The study was conducted in a French university hospital in Bordeaux. The population of interest was all types of healthcare workers, as vaccination was promoted for all hospital staff, regardless of patient relationships or working conditions. The corpus profiles were diversified as much as possible, considering socio-demographic characteristics such as age, gender and professional status. 

Two strategies were used to recruit participants: mobilization of the study team’s professional network; and voluntary solicitation on the way out of the staff restaurant. To approach new contacts, we asked study participants to refer potential volunteers.

### 2.4. Data Collection

The individual semi-structured interviews were conducted by one or two investigators trained in qualitative methods (CB, AB), either face-to-face at the hospital in a quiet and confidential place or by telephone, depending on the respondent’s preferences. Interviews were recorded, transcribed, and anonymized after oral consent had been obtained. 

The interview guide was based on the determinants of the comprehensive behavioral model of pandemic vaccination. Exploratory interviews were conducted to test and improve the guide. The themes addressed during the interviews were centered on (1) general representations of vaccination; (2) knowledge of and attitudes to COVID-19 vaccines; (3) COVID-19 vaccination intention; (4) expectations and obstacles associated with the vaccination campaign; (5) influence of peers and context. Examples of questions are presented in Table 1. The questions, initially in French, were translated by two investigators and checked by two native English speakers.

### 2.5. Data Processing and Analysis

All interviews were analyzed using a thematic approach. The analysis grid was constructed from several floating readings over time (Appendix A). New themes emerged during the analysis phase and were discussed by co-authors to ensure consistency. Participants were recruited until data saturation was reached and no more new themes emerged.

Data were coded by two researchers independently. An initial vertical analysis was performed by coding and analyzing the interviews individually. Then, the interviews were crossed for a horizontal analysis [27]. A final analysis for each professional category was conducted by: medical professionals (including physicians); para-medical professionals (including nurses, senior nurses, physiotherapists and healthcare assistants); and hospital technical professionals (including porters, administrative and technical staff). Each theme was summarized and illustrated with quotes. Data were discussed until consensus.

## 3. Results

### 3.1. Description of the Population

In total, twenty-five semi-structured interviews were conducted with HCWs from January to April 2021. Table 2 presents detailed characteristics of the participants. 

### 3.2. Determinants of COVID-19 Vaccination Intention

Our study revealed four main determinants of COVID-19 vaccination intention: (1) confidence in the vaccines being offered; (2) past experiences with vaccines and disease; (3) the opinions and vaccination status of others; (4) media handling of information related to vaccination against COVID-19. Each determinant is shown with quotes in Appendix A. A timeline is available in Appendix A and can be used to identify interview dates in relation to the different stages of the French vaccination campaign.

#### 3.2.1. Confidence in the Vaccines Being Offered

Given the pandemic context, the international scientific community mobilized to propose vaccines against COVID-19 as soon as possible. Questions were raised among HCWs about how fast these vaccines had been developed. Concerns were mainly focused on the efficacy and safety of the vaccines. One technical team member pointed out: *“It’s a vaccine that is still young, we still don’t have much information about the side effects”* (D1).

HCWs trust in new vaccines was very low. This was apparent when they criticized health authorities and pharmaceutical companies. They wondered what the true objectives of the vaccination campaign were; several HCWs believed that financial issues were more important than the health of the population. The rapid release of vaccines was associated with lobbying, and laboratories were believed not to care about the quality of the product. A physiotherapist emphasized: *“The fact that it’s fast (…) we are still suspicious (…) we don’t really know what is at stake (…) if it is really to protect people or to make money…”* (G2). Accordingly, when they did not trust the vaccines, HCWs appeared to be more reluctant to be vaccinated.

Moreover, a lack of trust in government authorities reinforced any mistrust of the proposed COVID-19 vaccines. This lack of confidence was found with regard to both vaccines and political authorities. It was illustrated by criticism and distrust in the decisions taken at the beginning of the vaccination campaign. For example, HCWs were suspicious that the campaign had begun by vaccinating the elderly with a product that they perceived to be unsafe and dangerous. A healthcare assistant said: *“I think the COVID-19 vaccine roll-out has been really fast (...) you get the impression that the elderly are being used as guinea pigs”* (A1).

#### 3.2.2. Experience Effect

The second determinant of COVID-19 vaccination intention was the level of proximity to COVID-19 disease and past experiences with various vaccines. Proximity to the disease can be defined as having been confronted with COVID-19 disease, deaths of relatives or patients, or having been ill. These experiences could influence HCWs’ vaccination intention either positively or negatively. Due to their professional activity, HCWs are directly confronted with the COVID-19 virus. This exposure raises awareness of the severity of the disease and positively influences HCW vaccination intention. A healthcare assistant said: *“I’ve lost old people in my family in the last year (...) I think you see things differently when you’re in it (...) I got COVID-19. I saw people die one by one, it’s really scary”* (A5).

However, there was some ambivalence about the perception of risk among HCWs. Working in the hospital and being more exposed to the virus can also create a feeling of overprotection. Some HCWs said that they were no longer afraid of getting the virus. A nurse testified: *“I think I’ve been exposed to it, now I’m not really afraid of getting the virus. I think I’ve either had it already or I’m immune”* (B4). When this is the case, vaccination intention may decrease, or vaccination can be delayed.

COVID-19 vaccination intention is also conditioned by experiences with other vaccinations. Interviews showed that a negative experience with another vaccination led to reluctance regarding COVID-19 vaccination. These experiences may have occurred in the private sphere or the professional sphere. Some HCWs expressed concern about side effects such as fever, which they may have had or observed after other vaccinations. A healthcare assistant who was unwell after the influenza vaccine recounted: *“Twice in a row I got a flu vaccination... I was sick!”.* This negative experience conditioned their COVID-19 vaccination intention: *“I am against the flu vaccine, so I am against the COVID vaccine”* (A2).

Physicians stood out for this determinant, as they considered that side effects are to be expected. An emergency physician claimed: *“There were no questions in my mind about the side effects, since that’s what a vaccine is all about (...) I had no fears”* (C4).

#### 3.2.3. The Opinions and Immunization Status of Others

HCWs said they do not consider the opinions or the vaccination status of others, as immunization is perceived as a personal decision. However, HCWs appeared to be sensitive to the opinions and the vaccination behavior of people in their entourage. A healthcare assistant said: *“When they (her colleagues at work) have been vaccinated, they will tell us how they felt afterwards and how they feel now, and that can reassure us too”* (A1). HCWs seemed to be attentive to the feelings of others and to expect feedback. Depending on the feedback they receive, they will either be more likely to be vaccinated or this will reinforce the reluctance.

On a wider scale, seeing large-scale mobilization for vaccination leads to a ripple effect and a compliance effect among HCWs. For example, an increase in the percentage of people vaccinated promotes their vaccination intention. This increase is perceived as a collective effort that gives meaning to vaccination and legitimizes vaccines. A physiotherapist, who was already hesitant during the interview, said: *“If only a small percentage of the population was vaccinated, well, I wouldn’t go running to be vaccinated first!”* (G1).

Once again, physicians were the exception and tended not to wait for a general trend in COVID-19 vaccination. Vaccination seems to be a standard and systematic act for them, and they displayed a sense of professional duty to themselves and towards others.

#### 3.2.4. Media Influence on Public Opinion

Finally, the way the media handles information regarding vaccines influences HCWs’ vaccination intention. They were not satisfied with the presentation and handling of news regarding COVID-19 vaccination. They believed the information was always negative and poorly argued from a scientific point of view. A member of the administrative staff complained: *“I get angry in front of the TV (...) because some of what they are saying is wrong! (...) It is not always very credible information (...) One has the impression that they have a bone there, and then that they must gnaw at it until it’s finished!”* (E1).

How information is disseminated has an impact on public opinion, even though HCWs report being aware that the news is often unreliable or inadequate. HCWs reflect an anxiety-inducing atmosphere around COVID-19 vaccination, which may, in turn, have an impact on COVID-19 vaccination intention. A porter said: *“It’s the current atmosphere in fact... We hear everything and its opposite... And it’s true that it leaves an uncertainty in our minds”* (F1).

This discrediting of vaccines leads to reluctance and doubt. Some HCWs preferred to postpone vaccination or to wait for other COVID-19 vaccines that they consider safer to become available. A care assistant said: *“I don’t think I will do it (COVID-19 vaccination) to begin with. Because we hear so much about anything and everything... I would prefer to wait a little and not rush”* (A4). A nurse also said: *“If in two or three years they found something or if there was no RNA in it, why not? But as long as there is RNA in it, there is no way I’m having it!”* (B3). These expressions of reluctance were less present among HCWs who use other sources of information, such as scientific articles.

## 4. Discussion

We found that the COVID-19 vaccination intention of French HCWs was essentially determined by their level of confidence in the proposed vaccines and personal or proxy experiences of the disease, the virus or vaccination. It was also influenced by the perception of the information provided by the media about COVID-19 vaccination and finally the opinion and vaccination status of others in the HCW environment.

The four determinants of HCWs’ vaccination intention identified were consistent with the determinants identified by the comprehensive behavioral model of pandemic vaccination in the general population [16]. Several studies also identify similar determinants in HCWs as in the general population [28,29,30,31,32]. We noted that the vaccination intention of HCWs seemed to be influenced by the perceived vaccination norm, perceived control, habits and experiences, level of knowledge, and confidence in health and political authorities. Like the general population, HCWs needed to be reassured and better informed about COVID-19 vaccination. Mandatory vaccination introduced in France for HCWs in October 2021 raised questions. This coercive measure is likely to reinforce distrustful attitudes and vaccine hesitancy [33]. Moreover, this measure will contribute to understaffing at a time when the need for HCWs is highest, both for COVID-19 patient care and in order to catch up on cancelled procedures and treat winter pathologies.

Vaccination intention depends on confidence in vaccines but also on political and health authorities. The SAGE group has already defined vaccine confidence as confidence in the safety and efficacy of vaccines, as well as in the system that provides them and in the motivations of the political authorities [8]. Confidence is in part based on high efficacy and few short- and long-term side effects [17]. However, vaccine confidence is part of a wider environment than the simple modalities of vaccines. It is conditioned by many contextual elements, with a link between vaccination intention, confidence in the vaccine and confidence in political and health authorities [34]. In France, confidence in vaccines has already been weakened by previous scandals [35]. Controversies over AstraZeneca’s vaccine (Vaxzevria) have also reinforced vaccine reluctance [36].

The reactive and restrictive management of the health crisis may also have affected vaccine confidence. In France, as elsewhere in the world, restrictive measures were taken to manage the epidemic [37]. These measures may have generated resistance and made vaccination a political issue. With the French crisis in vaccine confidence [38], it is important to promote expertise and consensus among HCWs and vaccination experts, as they are a more reliable source of information and deliver a more effective message about the benefits of vaccination [39,40,41].

The results showed that threat perception, along with fear of COVID-19, contributed to influencing vaccination intention. HCWs expressed the fact that they no longer feared the disease because they had been exposed to it daily without getting sick. This low perception of risk contributed to a decrease in vaccination intention [42,43,44]. The altruistic protection argument has proved to be effective in influencing HCW vaccination, but as for the influenza vaccine, there is no scientific consensus about the indirect protection of these vaccines [45,46]. From an individual perspective, highlighting the motivations for vaccination, such as self-protection from severe forms and protection of the most vulnerable, is recommended. This has already been demonstrated in the context of several vaccine controversies, such as vaccination against hepatitis B, in France [47,48]. It may also be important to promote positive vaccine experiences by amplifying the feedback of those who have been vaccinated. Transparency about the incidence of side effects to HCWs is important, especially during controversies such as that associated with the Vaxzevria vaccine [36].

HCWs mentioned media such as television and print as the most accessible and widely used sources of information. It is known that the media can be the source of rapidly spreading unreliable information [49], and misinformation has a negative impact on willingness to be vaccinated [50]. The literature also shows that social networks and the web have an impact on public opinion on vaccination [51]. Several studies have highlighted the predominance and visibility of anti-vaccine messages on these channels of information [52,53]. The link between misinformation and vaccine hesitancy might be part of the reason for health care workers’ reluctance to vaccinate. The media should deliver clear and positive messages, scientifically supported and also tailored to the target audience, local contexts, and vaccine characteristics [16]. Clear communication promotes better understanding and appropriation of information [54]. The mobilization of credible personalities is an opportunity that could be further exploited.

At the beginning of the COVID-19 vaccination campaign, health professionals were prioritized for vaccination [4]. French political and health authorities presented vaccination as a personal choice but enforced mandatory vaccination of HCWs eight months later [55]. Similarly, several HCWs in our study defended individual decision-making but also stated that they would be more in favor of being vaccinated if other people were. There is a contradiction in the discourse of both sides that must be called into question. The “free will” argument seems to be defended as an alternative that gives HCWs time to form an opinion about vaccines in a context that they do not find reassuring. However, the importance of their vaccination was widely publicized, and they were called upon to be vaccinated to protect others as part of their duty as caregivers. This puts them in an awkward position, caught between making a personal decision and their professional duty. Thus, some HCWs, referring to the vaccination behavior of others, showed sensitivity to the vaccine norm [56]. They were more inclined to adopt a behavior if it was “increasingly” being adopted by others [57,58]. It seems important to value the act of vaccination among HCWs in order to motivate and reassure others. Targeting and promoting their vaccination also encourages general vaccination intention through a compliance effect, observed in other vaccination contexts [59,60,61].

The analysis of the determinants of COVID-19 vaccination intention by professional category showed that physicians stood out. Studies have already shown differences in vaccine acceptance by professional category [42,62]; physicians expressed a high level of confidence in COVID-19 vaccines and in the scientific community. This may be due to greater interest in vaccination, easier access to information and therefore, greater distance from the information relayed by the mainstream media. People with a higher level of health literacy will tend to seek out scientific information at the source and are better trained to fact-check information in the media blur. There is a widening gap between those who have the literacy to seek out reliable information and those who do not. While many participants expressed distrust of media information, not everyone has the opportunity or the resources to form an informed and scientifically supported opinion.

The study has some limitations. The HCWs were recruited from a specific geographical area that was less affected by the epidemic than certain other regions of France [63]. In addition, the interviews were conducted in a rapidly changing context. The French vaccination campaign has undergone rapid and frequent changes in terms of the priority population, allocation and availability of vaccines. The results of the study are linked to an evolving pandemic context, which limits the generalization of data over time to a more stable and less restrictive context. However, conducting our study before the introduction of mandatory vaccination for HCWs certainly allowed for better identification of the determinants of vaccination intention. To our knowledge, few qualitative studies have contributed to the identification of the determinants of vaccination intention in the context of the COVID-19 pandemic among a key population such as HCWs [64,65]. Our study highlights the influence of the global environment on vaccination intention and underlines the need to implement systemic measures adapted to local specificities.

## 5. Conclusions

This study has highlighted the key determinants of COVID-19 vaccination intention among French HCWs, an important but understudied group. Their vaccination intention was thus influenced by confidence in the vaccines, past experiences with vaccines and disease, opinions and vaccination status of others, and media handling of information. It is important that these determinants are considered to mobilize effective levers for future vaccination campaigns. Our results can provide research directions on health communication in times of crisis, reinforcement of prosocial norms, and the use of an evidence-based approach. They can also guide preventive health policies and future vaccination campaigns to be adapted to local specificities and environments.

## Figures and Tables

**Figure 1 vaccines-10-01661-f001:**
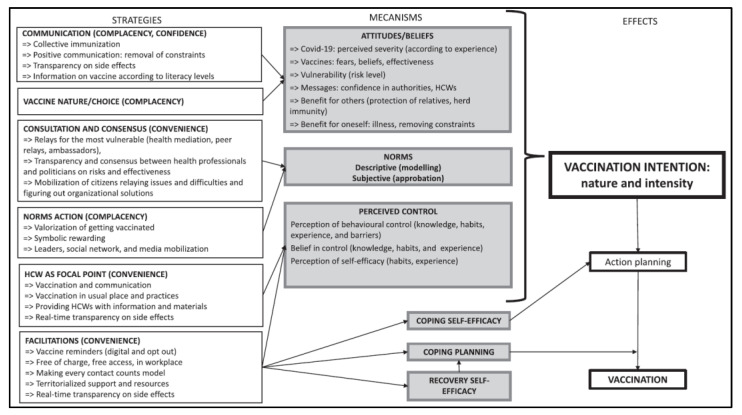
Meta-model: strategies to overcome COVID-19 vaccine hesitancy [16].

**Table 1 vaccines-10-01661-t001:** Semi-structured interview guide for health professionals at the University Hospital of Bordeaux, France, from January to April 2021.

Topics	Example of Questions and Reminders
General representations of vaccination	What do you think of vaccination in general? What is your relationship with vaccination in general? What about the influenza vaccine?
Knowledge of and attitudes to the available COVID-19 vaccines	What do you think of the COVID-19 vaccines being offered? How do you relate to these vaccines?
Vaccination intention for COVID-19 vaccines	Do you plan to get vaccinated in the coming months? Why/why not? What are the reasons, the advantages? If negative vaccination intention: What would make you change your mind?
Expectations of and obstacles to vaccination campaign	Could you tell me about the organization of the vaccination campaign? What are your expectations regarding the organization of this campaign?
Influence of peers and the context	What do your family/friends/work colleagues think about this vaccination? What does the vaccination of others mean to you? Can you tell me about your role in this vaccination campaign? What approach do you take with those around you? And with your patients?

**Table 2 vaccines-10-01661-t002:** Characteristics of participating health care workers, from January to April 2021, in Bordeaux University Hospital of France (N = 25).

Characteristics		N (%)
**Sex**		
	Men	9 (36)
	Women	16 (64)
**Age**		
	Under 50 years old	16 (64)
	Over 50 years old	5 (20)
	Unknown	4 (16)
**Profession**		
	Physician	4 (16)
	Nurse	4 (16)
	Healthcare assistant	6 (24)
	Physiotherapist	2 (8)
	Porter	2 (8)
	Senior nurse	1 (4)
	Administrative staff	3 (12)
	Medico-technical and technical staff	3 (12)
**Total**		**25**

## Data Availability

The data presented in this study are available on request from the corresponding author.

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
