# Peer review of "Determinants of COVID-19 Vaccination Intention among Health Care Workers in France: A Qualitative Study"

_vaccines, 2022, doi:10.3390/vaccines10101661_

Round 1
Reviewer 1 Report (Previous Reviewer 1)
This manuscript is underpowered. It only has 25 HCWs and based on that authors can not derive any statistical inference. This number can not represent the whole HCWs population in France. This study is not novel and was conducted more than a year ago. There are already other highly powered and multi center studies published from France that explored COVID-19 vaccine intention among HCWs. I do not believe this study adds anything beyond those already published and due to very small sample size, is not in a position to provide any conclusive information.
Similar and better powered articles:
1. https://pubmed.ncbi.nlm.nih.gov/33259883/
2. https://pubmed.ncbi.nlm.nih.gov/33964486/
3. https://www.mdpi.com/2076-393X/9/6/547
Author Response
The comments and suggestions of the first reviewer are not relevant to the qualitative method used for this article. Comments about the number of participants and statistics are not consistent with the method used. The three studies cited as similar to ours used quantitative methods. The point of a qualitative study is not to be representative of the target population, it is to have a better understanding of their motives and levers we can use. We take care to formulate our conclusion accordingly.
Therefore, we cannot modify our manuscript on the base of the first reviewer advice.
Reviewer 2 Report (New Reviewer)
Dear Editor
Thanks a lot for giving me the chance for reviewing this interesting and well written manuscript.
I have some minor comments;
Please review the conclusions since they are very basic and the section start with introductory sentences not conclusion sentences. Please re-construct the conclusion section focusing on the study outcome and show future direction.
I assume the interviews and the questions were prepared in French. If that correct, please indicate in the Methods how the translation was done, for example by independently by two authors or by professional service. This will help others to use this questionnaire in other languages.
Line 71 says “…the keys pillars (arrows in the figure) most likely…”. Figure 1 has many arrows, thin arrows and arrows as bullet points. Please indicate which arrow you exactly means (maybe by making them Bold) .
Line 92: please re-check the sentence in this line as it need spacing or correct punctuation.
For the other reviewer's comments on the Sample size: This is an "interview survey" not an "online survey". The example they provided with large number of participants are online survey. Usually in "interview survey" you get small number of participants and you donot do lots of statistical and p value calculations.
Author Response
We want to thank the second reviewer for his concerns and his advice. It was very helpful.
Following the second reviewer comment, the conclusion has been modified to synthesize the results of the study. Future directions have been developed, both for the research and for the actors of the vaccination campaigns. The methodological points mentioned in figure 1 (Meta-model: strategies to overcome COVID-19 vaccine hesitancy) and on the translation of the interview guide have been clarified. The punctuation errors (line 92) have been corrected.
Reviewer 3 Report (Previous Reviewer 3)
thanks
Author Response
Thank you fr your kind appreciation and for the advice in the first reviewing.
Reviewer 4 Report (Previous Reviewer 4)
1. Please develop and revise the manuscript according to the COREQ Statement.
2. It is suggested that COREQ Checklist completed and submitted as supplement with revised version of the manuscript
Author Response
The COREQ guidelines were used to complete the manuscript following the first review. At first we use the SRQR guideline (1) to write our article so we have modified our manuscript accordingly to this new guideline. The COREQ checklist can be found as a supplement to the revised version of the manuscript.
This manuscript is a resubmission of an earlier submission. The following is a list of the peer review reports and author responses from that submission.
Round 1
Reviewer 1 Report
This study is highly underpowered, with only 25 participants, and cannot reflect the healthcare population in France. Further, this study was conducted one year ago. With such a low number of participants and study performed one year ago, it cannot provide sufficiently useful information. There are not any figures or tables in the result section. All the results are descriptive without any data or statistics. It is understandable that with only 25 participants getting such information and using univariate and multivariate analysis are not possible. There are already several studies, with robust number of participants, published from France, e.g.:
1. https://pubmed.ncbi.nlm.nih.gov/33964486/
2. https://www.mdpi.com/2076-393X/9/6/547
Reviewer 2 Report
The manuscript provides an interesting insight into the knowledge of COVID-19 vaccination intention determinants among health care workers in France. The paper is clear and well-written. Data provided are sound. However, some minor revisions are required.
Comments/remarks:
Abstract
Subheadings are not allowed in this journal.
Introduction
I think that, in the introduction, it would be appropriate to give a brief definition of what is meant and which tasks are referred to when talking about Health Care Workers (HCWs).
In addition, the fact that COVID-19 vaccines were developed so fast and using relatively new techniques may have worsened vaccine hesitancy (DOI: 10.1186/s13584-021-00481-x). This should be added.
Methods
The article speaks about HCW in general, but when it comes to the study population, in the paragraph "Corpus recruitment", a distinction is made between healthcare workers and administrative and technical staff (lines 74-75).
Next, in line 100-101, it says that "a final analysis for each professional category was conducted: physicians, nurses, caregivers and the others HCWs". Here it appears that technicians and administrators are considered among the HCWs. Also in table 2 (Characteristics of participating health care workers), in the results section, there are porters, administrators, etc. included in the HCWs.
Results
What was the response rate? Did you perform a sample size calculation?
Line 120, “One hospital worker pointed out”: what kind of hospital worker? I think it is used as a synonym for HCW, however in the following citations the interviewee's job is always specified, perhaps it would be appropriate to report it here as well.
Discussion and conclusion
Line 214-215. Other studies have investigated how the perception of vaccine safety and effectiveness influences vaccine hesitancy (DOI: 10.3390/vaccines9111292, DOI: 10.1016/j.actpsy.2022.103550, DOI: 10.17179/excli2021-4439).
Line 309-310 “Intention to be vaccinated against COVID-19 among French HCWs hardly increased at the beginning of the mass vaccination campaign”: how can this be deduced from this study? If it is studied in literature, bibliographic reference is missing.
Reviewer 3 Report
Based on the review, I would like to suggest some Minor Revisions so that the worth of this manuscript can be enhanced. The necessary requirements are given as follows:
1) The paper has current references and demonstrate relevant literature. However, I missed several articles/papers (listed below). In this sense, some related papers can be added to the literature review:
https://doi.org/10.1590/0102-311x00115320
https://doi.org/10.1016/j.envres.2020.110148
https://doi.org/10.1371/journal.pone.0238214
https://doi.org/10.1016/S2214-109X(20)30300-4
https://doi.org/10.1080/15275922.2021.1940380
2) Introduction and case study must be improved.
3) Please extend further future research directions, and what about new and emerging application areas that will encourage and inspire future work on your field of research and application.
This study, I recommend the publication after minor revision.
Reviewer 4 Report
The manuscript has been generally good developed and presented. It is suggested that Authors employ COREQ statement in providing and presenting of the manuscript